# Analysis of NIH K99/R00 awards and the career progression of awardees

Nicole C Woitowich[1†], Sarah R Hengel[2†], Christopher Solis[3], Tauras P Vilgalys[4], Joel Babdor[5], Daniel J Tyrrell[6]*

[1]Department of Medical Social Sciences, Feinberg School of Medicine, Northwestern University, Chicago, United States; [2]Department of Biology, Tufts University, Medford, United States; [3]Department of Health, Nutrition, and Food Sciences, Florida State University, Tallahassee, United States; [4]Section of Genetic Medicine, Department of Medicine, University of Chicago, Chicago, United States; [5]Department of Systems Pharmacology and Translational Therapeutics, University of Pennsylvania, Philadelphia, United States; [6]Division of Molecular and Cellular Pathology, Department of Pathology, Heersink School of Medicine, University of Alabama at Birmingham, Birmingham, United States

*For correspondence:
danieltyrrell@uabmc.edu

[†]These authors contributed
equally to this work

Competing interest: The authors
declare that no competing
interests exist.

Reviewing Editor: Peter
Rodgers, eLife, United Kingdom

## Abstract

Many postdoctoral fellows and scholars who hope to secure tenure-track faculty positions in the United States apply to the National Institutes of Health (NIH) for a Pathway to Independence Award. This award has two phases (K99 and R00) and provides funding for up to 5 years. Using NIH data for the period 2006–2022, we report that ~230 K99 awards were made every year, representing up to ~$250 million annual investment. About 40% of K99 awardees were women and ~89% of K99 awardees went on to receive an R00 award annually. Institutions with the most NIH funding produced the most recipients of K99 awards and recruited the most recipients of R00 awards. The time between a researcher starting an R00 award and receiving a major NIH award (such as an R01) ranged between 4.6 and 7.4 years, and was significantly longer for women, for those who remained at their home institution, and for those hired by an institution that was not one of the 25 institutions with the most NIH funding. Shockingly, there has yet to be a K99 awardee at a historically Black college or university. We go on to show how K99 awardees flow to faculty positions, and to identify various factors that influence the future success of individual researchers and, therefore, also influence the composition of biomedical faculty at universities in the United States.

## eLife assessment

This study follows the career trajectories of the winners of an early-career funding award in the United States, and finds that researchers with greater mobility, men, and those hired at well-funded institutions experience greater subsequent funding success. Using data on K99/R00 awards from the National Institutes of Health's grants management database, the authors provide **compelling** evidence documenting the inequalities that shape faculty funding opportunities and career pathways, and show that these inequalities disproportionately impact women and faculty working at particular institutions, including historically black colleges and universities. Overall, the article is an **important** addition to the literature examining inequality in biomedical research in the United States.

## Introduction

Postdoctoral research fellows are a driving force of the academic biomedical research enterprise. They typically have doctorate (i.e., PhD) or medical degrees (i.e., MD or DVM) which means they have completed a significant amount of education, training, and publishing (*Wright and Vanderford,*

2017; *Igami et al., 2015*; *Ghaffarzadegan et al., 2015*; *Larson et al., 2019*). Postdoctoral fellows aspiring to continue in academic research typically seek tenure-track faculty appointments (*Larson et al., 2019*). Despite recent trends of new doctoral degree holders seeking private sector positions (*Heggeness et al., 2017*), tenure-track academic faculty positions are highly coveted and competitive due to the increasing number of PhD graduates but stagnant number of tenure-track faculty positions (*Ghaffarzadegan et al., 2015*; *Daniels, 2015*). Obtaining such positions requires consistent access to opportunities for success such as attending doctoral programs at perceived prestigious institutions, publishing manuscripts at the undergraduate, graduate, and postdoctoral levels, and receiving extramural grant funding throughout one's career (*Clauset et al., 2015*; *Fernandes et al., 2020*; *Brechelmacher et al., 2015*; *van Dijk et al., 2014*). University prestige has significant impacts on resources (*Way et al., 2019*), paper acceptance rates (*Okike et al., 2016*), citations (*Crane, 1965*), and awards (*Schlagberger et al., 2016*). There is also significant bias against women, LGBTQIA+, and other systematically marginalized groups at all levels, which results in underrepresentation in STEM fields and fewer R01 awards compared to white/men counterparts (*Wapman et al., 2022*; *Safdar et al., 2021*; *Oliveira et al., 2019*; *Check Hayden, 2015*). To combat these biases, the National Institutes of Health (NIH) developed funding mechanisms, such as the K99 MOSAIC and K12 IRACDA awards, to promote scientists from diverse backgrounds into tenure-track faculty positions. In addition, other non-federal grant mechanisms exist, such as the Gilliam Fellowships for Advanced Study from the Howard Hughes Medical Institute which began in 2004. The success of these programs at addressing bias or improving equity for systematically marginalized groups in the tenure-track ranks is currently unclear because the programs are either too new (MOSAIC) or data are not publicly available (HHMI Gilliam Fellowships).

Within the United States, one of the most lucrative award mechanisms to facilitate the transition from postdoctoral fellow to tenure-track faculty member in the biomedical sciences is the NIH Pathway to Independence Award (K99/R00) (*Carlson et al., 2016*). Launching in 2006, this award provides salary support for postdoctoral fellows for 2 years during the K99 phase and $250,000/yr for 3 years during the R00 phase (*Report, 2024*; *Funding, 2017*). The candidate benefits directly through funding, visibility, and protected time (*Wright and Vanderford, 2017*; *Funding, 2017*). Having a K99/R00 award also demonstrates to hiring institutions that the candidate has a track record of receiving extramural funding which is a requirement for future success in academic research careers (*Hsu et al., 2021*). Roughly 89% of K99 awardees received R00 funding on a rolling yearly basis, which indicates having obtained a faculty position although factors such as publications, presentations, interpersonal skills, choice of advisor, and others also contribute to securing faculty positions. Of those that secure faculty positions, many factors influence postdoctoral researchers in their decision of which position to choose including career-oriented factors like department, support, colleagues, and personal factors such as family and location. About 50% of R00 awardees subsequently attain R01 funding (*Pickett, 2019*) which is a higher success rate than other NIH career development grants such as K01, K08, and K23 (*Carlson et al., 2016*; *Conte and Omary, 2018*), although having any K-award tends to increase future funding success (*Nikaj and Lund, 2019*). The specific and cumulative factors that influence future NIH funding success for R00 awardees has not been examined.

Individuals from marginalized communities are typically at a disadvantage when applying for research grants such as R01s. For example, men typically experience 2–3% greater funding success rates; however, in 2019 women had an advantage in funding success rates for NIH research project grants (21% vs 20%) (*Chaudhary et al., 2021* ). Men also receive an average of $35,000–$45,000 more NIH funding per grant than women or other gender identities (*Oliveira et al., 2019*; *Chaudhary et al., 2021*; *Mayes et al., 2018*), however, some report no difference (*Pohlhaus et al., 2011*). Women have been reported to have a disadvantage in transitioning from K-awards to R-awards from the NIH (*Nguyen et al., 2023*). Racial and ethnic minorities are also at a disadvantage in receiving R01 funding compared to white counterparts (*Ginther, 2022*; *Ginther et al., 2011*). The institution itself also influences R01 funding success. Here, we explore the conversion of K99 to R00 awards by year, gender identity, and institutions (other demographic data are not publicly available) to demonstrate the flow of K99/R00 awardees from postdoc to faculty positions. We also determine whether these influence future NIH funding success for K99/R00 awardees. These findings have the potential to influence how and where career development awards are made, how potential candidates are supported, and how grant reviewing practices may be changed to be more equitable and inclusive.

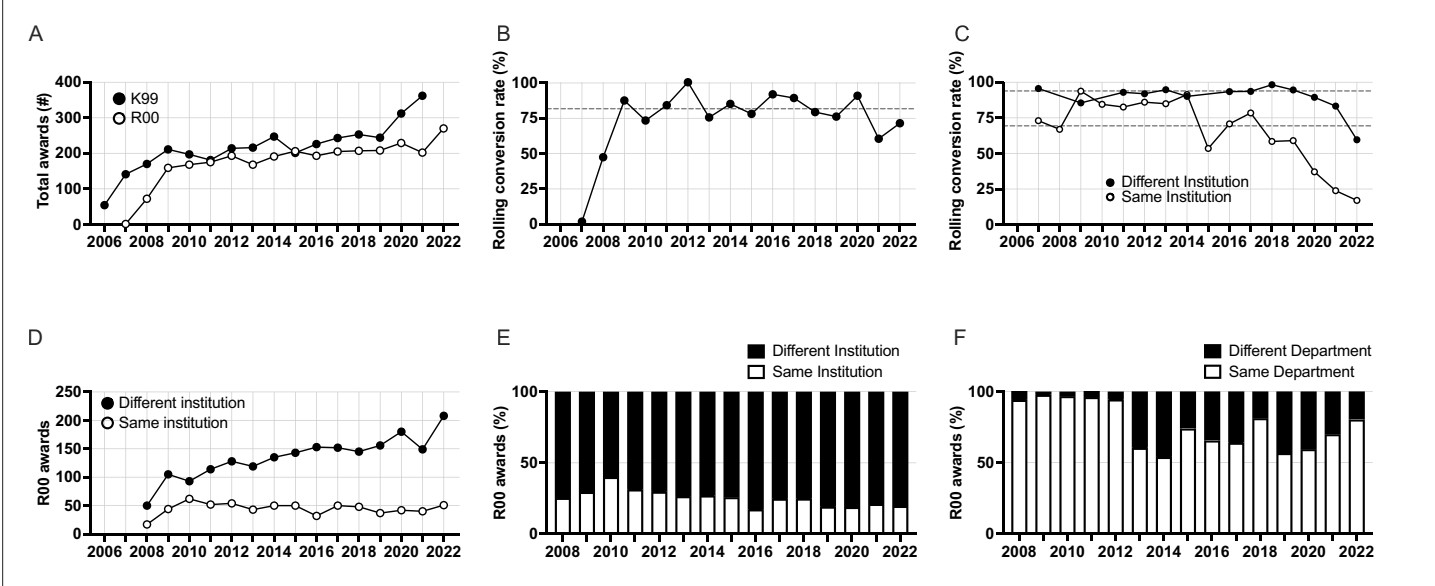

**Figure 1.** Rate of K99 and R00 awards and conversion by year. (**A**) Total extramural K99 awards and R00 awards made by year. (**B**) Annual rolling conversion rate of R00 award activation compared to the number of K99 awards activated in the prior year. The formula is: $(X/(Y − 1)) × 100$, where $X$ is the number of R00 awards made in a year and $Y$ is the number of K99 awards made in a year. Average rolling retention rate (81.8%) from 2008 to 2022 shown in dashed line. (**C**) Annual rolling conversion rate of R00 award activation compared to the number of K99 awards activated in the prior year stratified by whether the R00 was activated at the same or different institution as the K99 award. Average retention rates for R00 at different institution (93.9%) and the same institution (69.4%) are shown in dashed lines. (**D**) Total number of extramural R00 awards per year at the same institution or different institution. (**E**) Annual stacked bar plots showing the percentage of R00 awards and whether they were at the same institution or different institution. (**F**) Within the K99 to R00 conversions at the same institution, annual stacked bar plots showing the percentage of R00 awards and whether they were at the same department or different department.

The online version of this article includes the following source data for figure 1:

**Source data 1.** Source data for the number of K99 and R00 awards, and rolling conversion rate.

## Results

### K99 awardees increasingly move to other institutions for R00 awards from 2008 to 2022

K99 awards provide funding for 2 years and awardees generally can accept faculty positions within the second year and convert the K99 award to an R00 award. From the total of 3456 extramural K99s awarded from 2006 through 2021 (*Figure 1A*), 2842 extramural R00 awards have been made by the end of 2022 (*Figure 1A*). 141 of the R00 awardees do not have K99 data in the NIH Reporter database and thus must be intramural K99 awardees (i.e., awardees at NIH research laboratories). Since we do not have data on intramural K99 awardees who do not convert to the R00 award, we will exclude these 141 R00 awardees. There are 755 extramural K99 awardees that did not receive R00 awards. 426 of these received K99 awards in 2020 or after and may still be in the K99 phase; however, 329 were from 2019 or earlier, which suggests they have not and are no longer eligible for R00 awards. Where those awardees went and why they did not receive an R00 is unknown.

From 2008 through 2021, there were an average of 234 extramural K99 awards made per year, and from 2009 to 2022, an average of 198 R00 awards per year. 2701 extramural K99 awardees activated R00 awards, for a total of 78% conversion. On a rolling basis, excluding 2006–2007 (first 2 years), 81.8% of K99 awardees go on to receive an R00 award (*Figure 1B*). Until 2015, the rolling conversion rate was similar among R00 awardees who were at different institutions than the K99 award and R00 awardees at the same institution (*Figure 1C*). Starting in 2015, the activation rate of R00 awardees at the same institution decreased and has been lower than R00 awardees at different institutions since then (*Figure 1C*). The average rolling conversion rate for R00 awardees at the same institution was 69.4% whereas it was 93.9% for R00 awardees at different institutions. In the first 4 years of the K99/R00 program, around 30% of R00 awardees had been hired at the same institution;

however, from 2013 onward, the proportion of awardees that stayed at the same institution declined to ~20% and has remained consistent (*Figure 1D, E* ). The number of R00 awardees at the same institution was consistent from 2010 to 2022 but the number of R00 awardees at different institutions has increased over the same period (*Figure 1D, E*). In the beginning of the K99/R00 program, nearly all the awardees that stayed at the same institution stayed in the same department; however, from 2013 onward a greater number moved to new departments (*Figure 1F*). This is in line with what Pickett reports that ~20% of K99 recipients received their first R01 award at the same institution where they had a K99 award (*Pickett, 2019*). Thus, the number of K99 to R00 conversions is consistent over time, but increasingly more R00 awardees have moved to other institutions since 2013.

## Geographic localization of K99 awards are at a more concentrated set of institutions whereas R00 awards are more dispersed

K99 awards have been granted to individuals at 256 different institutions and R00 awardees represented 357 unique institutions. Strikingly, 54% of all K99 awardees were at 10% of all institutions, while 46% of all R00 awardees were at 10% of institutions. K99 and R00 awards were both most frequent in parts of the country containing a high concentration of research universities (e.g., the Bay Area in California and Boston Area in Massachusetts) (*Figure 2*). Three institutions had more than 75 K99 awardees through the end of 2022 (Stanford [237], University of California San Francisco [116], and Massachusetts General Hospital [93]), and 73 institutions had just one, with the median being four awards per institution. Remarkably, Stanford had more awards than the 134 institutions with the fewest number of awards (*Figure 2*). R00 awards were dispersed across more institutions than K99 awards, with the maximum number being 60: three institutions had more than 50 (Massachusetts General Hospital [60], University of Pittsburgh [58], and University of Michigan [54]), and 116 institutions had just one (*Figure 2*). K99 awards are more concentrated in East and West coast states than R00 awards (73% vs 55%). California (*n* = 491), Massachusetts (*n* = 348), and New York (*n* = 121) had the largest proportion of K99 relative to R00 awards, and Texas (*n* = 48), Ohio (*n* = 46), Michigan (*n* = 33), and Arizona (*n* = 31) received the greatest number of R00 awardees relative to K99 awards. These data indicate that while more K99 awards are physically located on the East and West coast universities, the R00 awardees are distributed more evenly throughout the geographic location of the United States.

## Institutions with the most NIH funding tend to hire K99/R00 awardees from other institutions with the most funding

We next analyzed the flow of awardees from K99 institution to R00 institution (*n* = 2703 individuals) (*Figure 3*). We coded the 25 institutions that received the most NIH funding in the year 2022 as '1' and the rest as '2'; however, Leidos Biomedical Research, Inc is included in the highest funded institutions but has no K99 or R00 awardees. The institutions that produce the highest number of K99 awardees also tend to hire the most R00 awardees and also tend to receive the most NIH funding (*Figure 3*). Most K99 awardees attain faculty positions at private institutions (62%) compared to public institutions (38%). Private institutions with the highest NIH funding generally hire R00 faculty from other private institutions whether they are in the top NIH funding category (31.2%) or not (46.9%). The public institutions with the highest NIH funding and private institutions that do not have the highest NIH funding also tend to hire R00 faculty more from private institutions categorized as both top NIH funding (20.3% and 22.5%, respectively) and lower NIH funding (27.2% and 41.3%, respectively). Notably, the public and private institutions with the largest NIH funding portfolios and other private institutions have hired only 26% of K99 awardees from public institutions with smaller NIH portfolios. In contrast, these public institutions with smaller NIH funding portfolios have hired 74% of all K99 awardees from similar institutions (*Table 1*).

## K99/R00 awardee self-hires are more common at institutions with the top NIH funding

We also demonstrate the number of self-hires of K99/R00 recipients from within institutions. A mean of 27% of K99 awardees remain at their home institution during the R00 phase; however, this differs based on institutional type and NIH funding portfolio size. At private institutions with large NIH funding portfolios, 36% of those stay at the same institution for the R00 phase. Fewer individuals stay at the same institution from public institutions with large NIH funding portfolios (30.9%), private

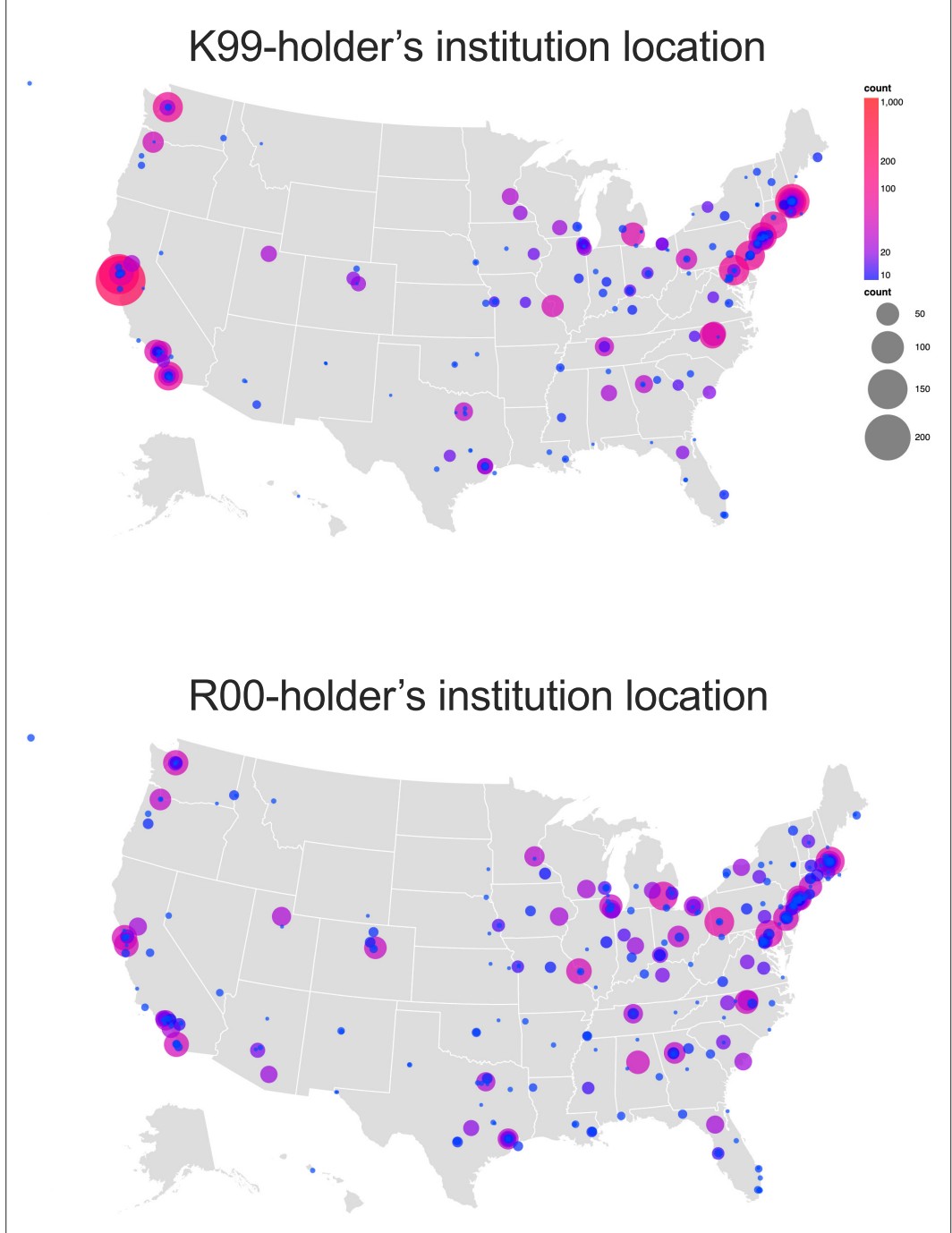

**Figure 2.** Cartographic representation of the number of K99 and R00 awards by institutional location and state from 2007 to 2008 through 2021–2022. Note no individual award or grant is counted more than 1 time.

The online version of this article includes the following source data for figure 2:

**Source data 1.** Source data containing the list of K99 and R00 awards and the location coordinates used to generate the map figure.

smaller NIH funding portfolios (23.3%), and public smaller NIH funding portfolios (18.5%). 70% of the R00 awardees recruited by Massachusetts General Hospital were self-hires. Of the remaining 17 individuals, 8 were from other Harvard-associated institutions (Brigham and Women's Hospital, Dana Farber Cancer Institute, Broad Institute). There are other institutions that have higher relative proportions of K99 awardees remain for the R00 phase (i.e., University of Pittsburgh, Johns Hopkins

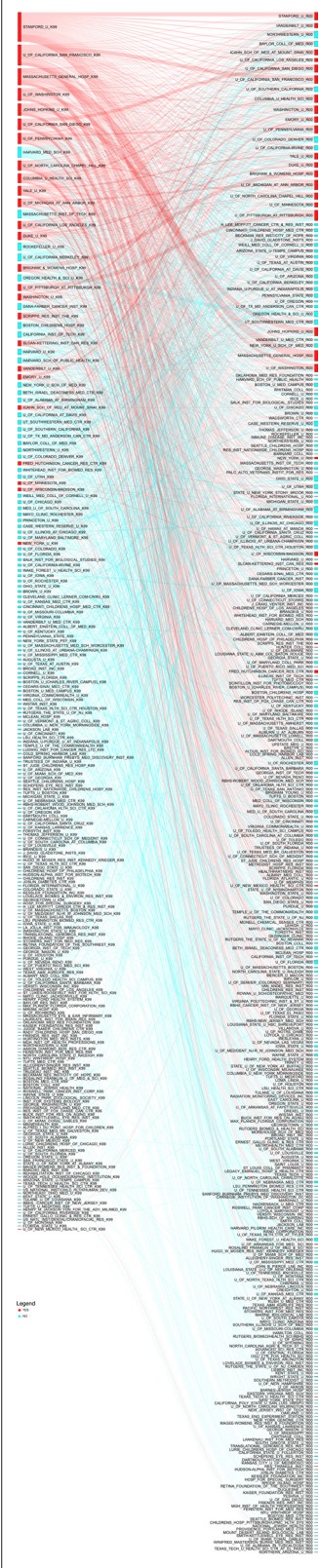

**Figure 3.** Sankey diagram of all successful K99 to R00 award transfers from 2007 to 2022 with the K99 institution on the left and R00 institution on the right. An interactive version of this figure can be found at:

*Figure 3 continued on next page*

Interactive Sankey diagram can be found at: https://k99tor00.shinyapps.io/K99-R00_Sankey/. Source data for **Figure 3** are in **Supplementary file 3**.

University, and Brigham and Women's Hospital). In contrast, some institutions recruit more of their R00 awardees from external institutions (i.e., Northwestern University, University of Texas Southwestern, University of Utah, and Yale University).

## More K99 awardees are men than women

We performed a name-to-gender classification using binary terms of man and women. This type of classification is inherently biased unevenly across demographics and groups (please see Methods) (*Lockhart et al., 2023*). Using this methodology, we found that men make up the majority of K99 (*n* = 2028, 58%) and R00 (*n* = 1655, 58%) awardees. The same percentages of men and women K99 awardees convert their K99 awards to R00 awards (*Table 2*). Since the beginning of the K99 award mechanism in 2006, men have received more K99 awards than women each year; however, the percentage of women receiving K99 awards has risen slightly in the most recent decade (*Figure 4*).

## K99 and R00 awards are concentrated within the highest funded institutions

The majority of K99 awardees were at private institutions with 28% from the top 25 highest funded institutions and 35% from other private institutions. The remaining 37% of K99 awardees were from public institutions (*Figure 5A*, *Tables 3 and 4*). We examined the composition of institutions represented by K99 awards (*Figure 5B*). We found that 45% of K99 awards were made to the top 25 highest funded institutions which represents only 24 institutions while 55% of K99 awards were made to 226 lower funded institutions (*Figure 5B*). The percent of K99 awards made at each institution type has remained consistent on an annual basis (*Figure 5C*). The majority of R00 awardees were at lower funded institutions with 38% at public institutions 38% and 29% at private (*Figure 5D*, *Tables 3 and 4*). Only 32% of R00 awards were made to the top 25 highest NIH-funded institutions. R00 awards were distributed slightly more evenly among institutions with 32% of R00 awards going to the top 25 highest funded institutions and 68% going to

**Table 1.** Number and frequency of R00 faculty hires from specific types of K99 institutions from 2006 to 2022.

Note, each column sums to 100%. *Excluding self-hires to the same institution.

| | Top NIH funding, private R00 n (%) | Top NIH funding, public R00 n (%) | Not top NIH funding, private R00 n (%) | Not top NIH funding, public R00 n (%) |
|---|---|---|---|---|
| Top NIH funding, private K99 | 295 (56.0%)<br>105 (31.2%)* | 71 (20.3%) | 179 (22.5%) | 196 (19.0%) |
| Top NIH funding, public K99 | 47 (8.9%) | 152 (43.5%)<br>44 (18.3%)* | 94 (11.8%) | 174 (16.9%) |
| Not top NIH funding, private K99 | 157 (29.8%) | 95 (27.2%) | 437 (55.0%)<br>252 (41.3%)* | 251 (24.3%) |
| Not top NIH funding, public K99 | 28 (5.3%) | 31 (8.9%) | 85 (10.7%) | 411 (39.8%)<br>220 (26.2%)* |
| Column total | 527 (100%) | 349 (100%) | 795 (100%) | 1032 (100%) |

lower funded institutions which represents 327 institutions (**Figure 5E**). This means that the top 25 highest funded institutions had an average of 65.5 K99 awards and 38.2 R00 awards per institution. In contrast, the lower funded institutions had an average of 8.4 K99 awards and 5.8 R00 awards per institution. The percent of R00 awards made at each institution type was also consistent on an annual basis (**Figure 5F**). While a greater percentage of both K99 and R00 awards are made at lower funded public and private institutions, the top 25 highest funded institutions clearly receive and retain a far greater share of both K99 and R00 awardees.

## Being female, staying at the same K99 and R00 institutions, and activating the R00 at a lower funded institution are disadvantages to future funding success

We sought to determine which factors impacted whether K99/R00 awardees received major subsequent extramural funding in the form of an NIH R01, DP2, or R35 MIRA grant. We refer to these as 'major awards' as they fund a similar total cost, timeframe, and hold similar weight for tenure and promotion considerations. Thus, we calculated a Cox proportional hazard model which includes whether the K99/R00 awardees stayed at the K99 institution for the R00 phase or left, were men or

**Table 2.** Number and frequency of men and women receiving K99 and R00 awards and whether they successfully converted K99 to R00 awards from 2006 to 2022.

| | N | % |
|---|---|---|
| K99 recipients | 3474 | 100 |
| Men | 2028 | 58 |
| Women | 1395 | 40 |
| Unknown | 51 | 1 |
| R00 recipients | 2843 | 100 |
| Men | 1655 | 58 |
| Women | 1142 | 40 |
| Unknown | 46 | 1 |
| Successful K99–R00 transition | 2703 | 100 |
| Men | 1575 | 58 |
| Women | 1090 | 40 |
| Unknown | 38 | 1 |

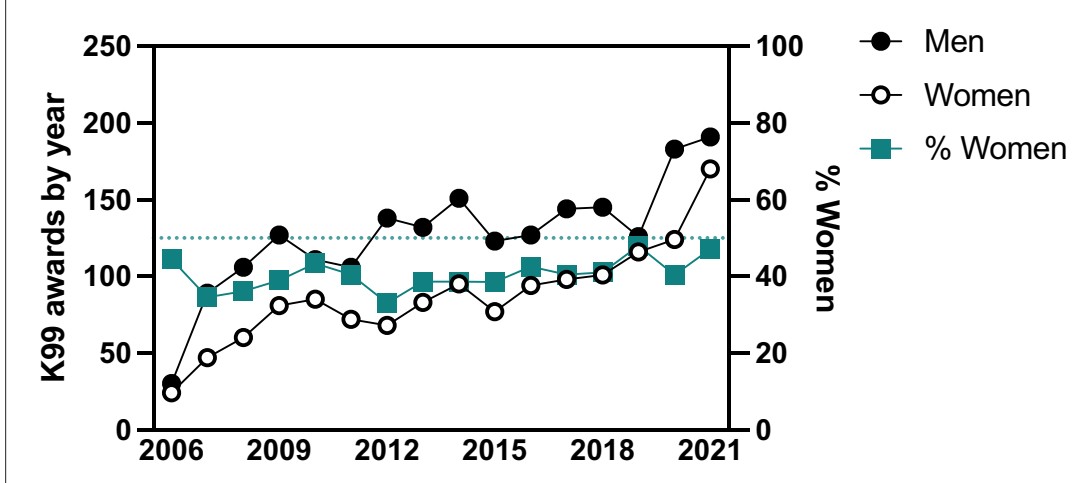

**Figure 4.** Number of K99 awards per year to either men or women (left *Y*-axis) and the percentage of K99 awardees that are women (right *Y*-axis).

The online version of this article includes the following source data for figure 4:

**Source data 1.** Source data of the number of K99 awards by year stratified by men or women.

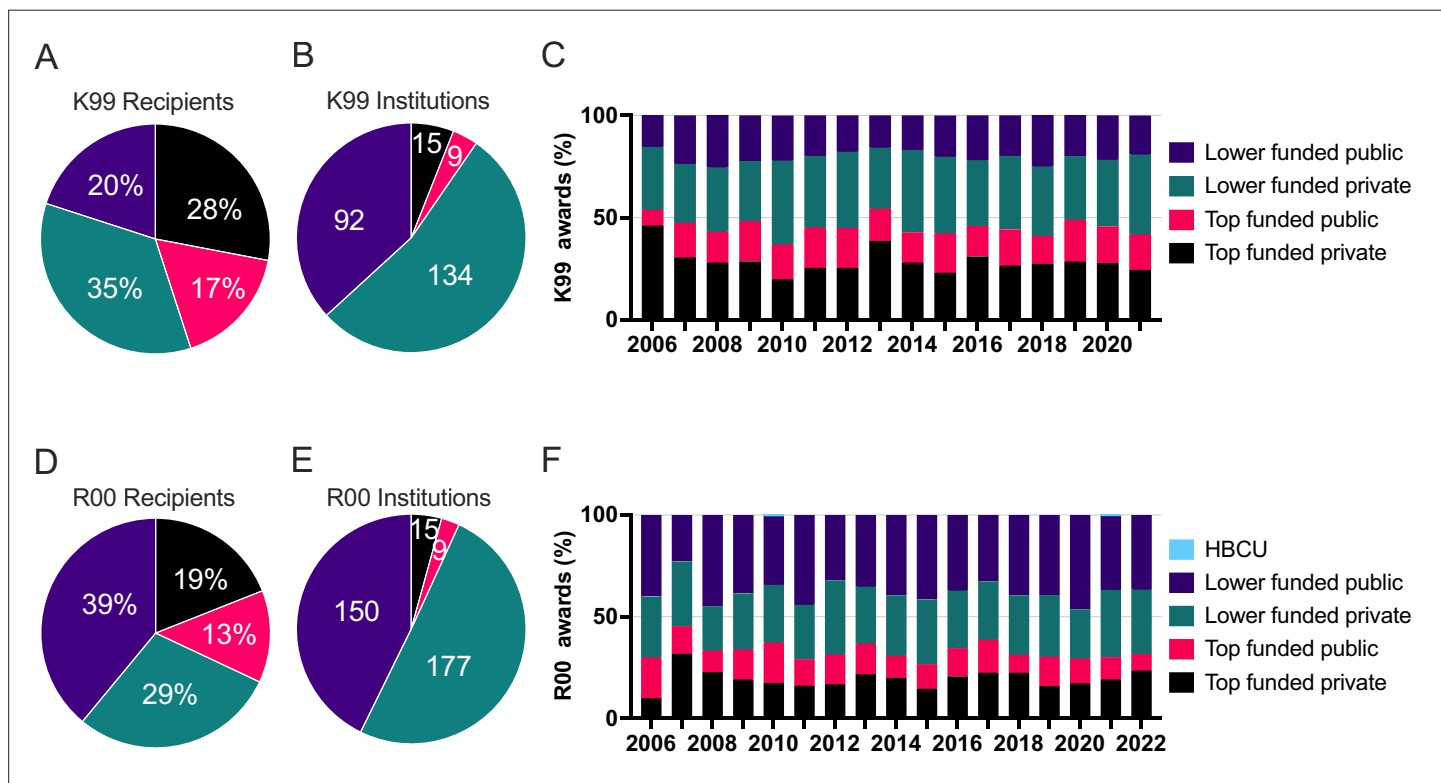

**Figure 5.** The percentage of recipients by the institution type for K99 (**A**) and R00 (**D**) and the number of different institutions within institution type that has received K99 (**B**) and R00 (**E**) awards from 2006 to 2022. The percentage of institution types receiving K99 (**C**) and R00 (**F**) awards by year from 2006 to 2022.

The online version of this article includes the following source data for figure 5:

**Source data 1.** Source data of the number of K99 and R00 awards made at each institution type per year.

**Table 3.** The number and frequency of K99 and R00 awardees by institution type. Successful K99–R00 transitions show the K99 award institution from 2006 to 2022. The 2842 R00 recipients includes recipients of National Institutes of Health (NIH) intramural K99 awards.

| | N | % |
|---|---|---|
| K99 recipient institution | 3473 | 100 |
| Top funded private | 990 | 28.5 |
| Top funded public | 582 | 16.8 |
| Lower funded private | 1212 | 34.9 |
| Lower funded public | 689 | 19.8 |
| R00 recipient institution | 2842 | 100 |
| Top funded private | 549 | 19.3 |
| Top funded public | 368 | 12.9 |
| Lower funded private | 831 | 29.2 |
| Lower funded public | 1094 | 38.5 |
| Successful K99–R00 transition K99 institution | 2703 | 100 |
| Top funded private | 741 | 27.4 |
| Top funded public | 467 | 17.3 |
| Lower funded private | 940 | 34.8 |
| Lower funded public | 555 | 20.5 |

women, whether the K99 or R00 institutions were private or public, and whether the K99 and R00 institutions were one of the top 25 institutions for NIH funding in 2022 (*Figure 6*). Individuals that stayed at the same institution for the K99 and R00 phases were at the greatest disadvantage for receiving future funding (hazard ratio: 0.7095, p < 0.0001). The next strongest factor in receiving future R01 funding was whether the R00 institution was in the top 25 largest NIH funding portfolios (p < 0.0001). The third factor was whether the candidate was a man or woman (p = 0.001), consistent with previous findings that women are at a disadvantage in academic career placement (*Pinheiro et al., 2017*). The NIH funding portfolio of the K99 institution and whether the K99 institution was private or public did not impact success in receiving future R01 funding.

The similarity in effect sizes suggests a cumulative advantage, or disadvantage, model. K99/R00 awardees that were men and moved to a new institution with large NIH funding portfolios for the

**Table 4.** Number and frequency of institution type that received K99 and R00 awardees from 2006 to 2022.

| | N | % |
|---|---|---|
| K99 recipient institution | 250 | 100 |
| Top funded private | 15 | 6.0 |
| Top funded public | 9 | 3.6 |
| Lower funded private | 134 | 53.6 |
| Lower funded public | 92 | 36.8 |
| R00 recipient institution | 351 | 100.0 |
| Top funded private | 15 | 4.3 |
| Top funded public | 9 | 2.6 |
| Lower funded private | 177 | 50.4 |
| Lower funded public | 150 | 42.7 |

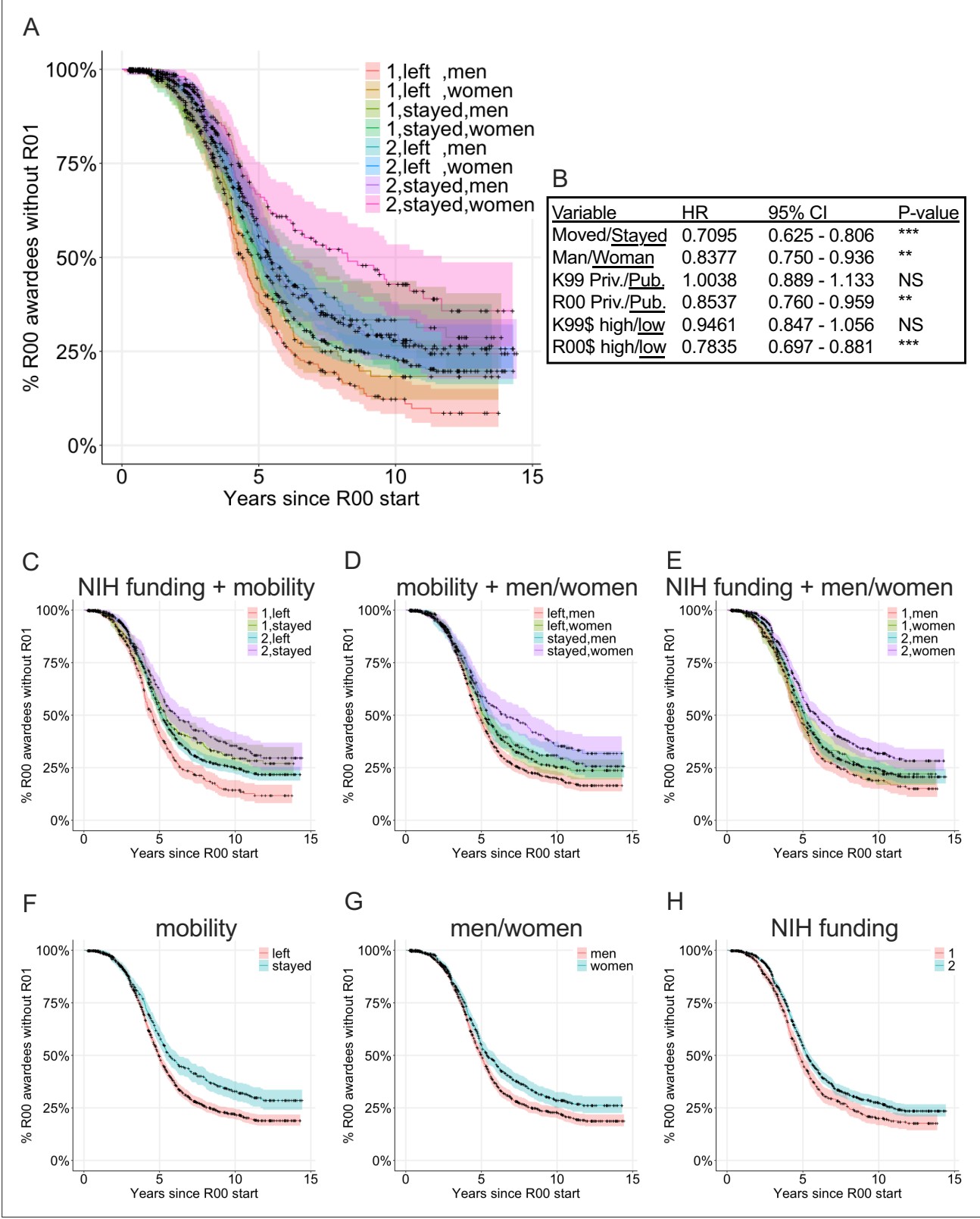

**Figure 6.** Cox proportional hazard model examining K99/R00 awardees success at receiving subsequent major extramural National Institutes of Health (NIH) awards by mobility, classified as women or men, and R00 institutional NIH funding level. Mobility is classified whether the K99 awardee moved to a new institution for the R00 award activation or stayed at the same institution. Classification of women or men was determined by name (see Methods for description and limitations). R00 institution funding level is classified as: 1: 25 highest NIH-funded institutions in 2022 and 2: all other institutions.

*Figure 6 continued on next page*

*Figure 6 continued*

(**A**) Survival curve demonstrating all eight possible classifications. (**B**) Cox proportional hazard model hazard ratio, 95% confidence interval, and p-value. Survival plots showing the individual components of the full model including (**C**) NIH funding + mobility, (**D**) mobility + women/men, (**E**) NIH funding + women/men, (**F**) mobility, (**G**) women/men, and (**H**) NIH funding.

The online version of this article includes the following source data for figure 6:

**Source data 1.** The source data used to calculate the hazard ratio including whether R00 awardees received a future award, were men or women, were at private or public K99 institution, were at private or public R00 institution, were at high or lower funded K99 institution, were at high or lower funded R00 institution, and whether they moved to a new institution for the R00 award phase.

R00 phase of the award have the shortest median time to receiving a subsequent major NIH award (median 4.6 years from the R00 budget start date to major award budget start date) as well as greater overall percent chance of ever receiving a major award (84.6%; *Figure 6*). Each cumulative disadvantage increased the median time to receive a major NIH award by about 6 months and reduced the overall chance to receive subsequent major NIH award by 5–6%. The median time to receive subsequent major NIH awards for K99/R00 awardees that were women that stayed at an R00 institution with smaller NIH funding portfolios is 7.4 years and there is only a 60% chance to receive any major NIH award. This amounts to 2.8 years longer and 24.6% lower chance to receive any subsequent major funding compared with men who move to institutions with the largest NIH funding portfolios for the R00 phase of the award. This length of time is critical for tenure-track faculty who typically have between 6 and 7 years to earn tenure which often hinges on receipt of a major research grant for research-intensive faculty.

## Faculty at HBCUs mostly have doctoral degrees from HBCUs and public and private institutions outside of the top 25 NIH funding category

It is clear that diversity in all metrics is beneficial for scientific progress. We sought to examine the rates of funding for those at historically under-funded institutions. We examined HBCUs in NIH Reporter, we found that there were eight active R01 awards at Morehouse School of Medicine, nine at Howard, five at Meharry Medical College, and five at Jackson State University in April 2023. Strikingly, of all 2703 K99 awardees that transitioned to R00 awards, no K99 awardees were from HBCUs, and only two K99 awardees activated R00 awards at HBCUs. Thus, there are not enough data available to determine how the classification of HBCU impacts future R01 funding success; however, Wapman et al. have made an extensive dataset of all United States faculty from 2010 to 2020 and where they received their doctoral degrees publicly available (*Wapman et al., 2022*). Wapman et al. did not demonstrate doctoral degrees or faculty hiring practices at HBCUs. This prompted us to examine faculty flow at HBCUs. We examine where faculty with doctoral degrees from HBCUs ended up for their faculty career (*Figure 7A*) and where faculty members at HBCUs received their doctoral degrees (*Figure 7B*) specifically related to biological and biomedical faculty. We included only biology, biomedical, and health-related fields (see Methods) which excluded fields such as computer science, business, language, etc., which resulted in a total of 65,120 faculty flows from doctoral degree programs.

Out of the 65,120 total faculty flows, there are data on 296 faculty members with doctoral degrees from HBCUs with a known doctoral degree granting institution. There were 529 faculty at HBCUs with a known doctoral degree granting institution. Those with doctorates from primarily underrepresented minority-serving institutions are less likely to be in academic positions (*Martinez et al., 2018*). The HBCU granting the most doctoral degrees that went on to receive faculty appointments in the United States were from Howard University with 116 individuals. The majority of these are working as faculty members at Howard (56). The HBCU with the next most doctoral degrees in faculty positions is Florida A&M with 44 and the bulk of these (32) are employed at Florida A&M. Most of the institutions with faculty that have doctoral degrees from HBCUs are other HBCUs and public and private institutions. Notably there are very few HBCU doctoral degree holders working at public or private institutions within the top 25 NIH funding category (*Figure 7A*). When examining faculty flows to HBCU's it is apparent that the HBCU faculty ranks employ a large number of faculty with HBCU doctoral degrees (*Figure 7B*). There are more faculty at Howard University with doctoral degrees from a range of different types of institutions which are primarily public and private and institutions outside of the top 25 NIH funding category; however, there are some (nine from University of Michigan, one from Vanderbilt, four from University of Wisconsin, and three from UNC). For the other HBCUs, the faculty ranks

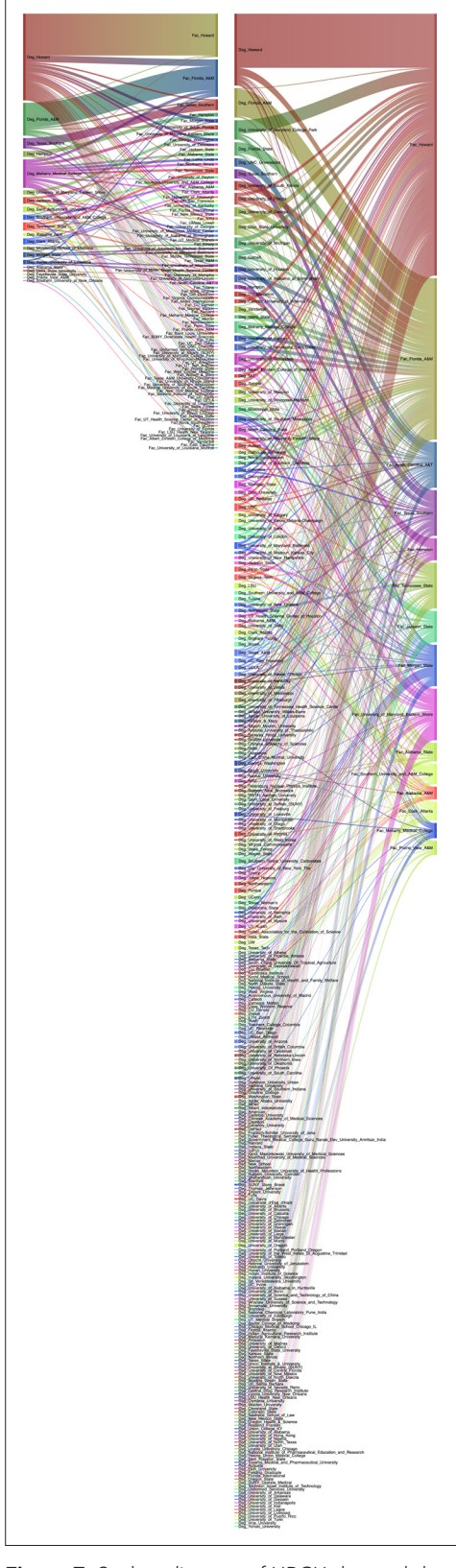

**Figure 7.** Sankey diagram of HBCU doctoral degrees and where those degree holders are faculty members and of HBCU faculty and where they received their doctoral degrees from. (**A**) Sankey diagram of faculty

*Figure 7 continued on next page*

*Figure 7 continued*

members that received doctoral degrees from HBCUs (on left) and the faculty institution they moved to (on right). (**B**) Sankey diagram of faculty members at HBCUs (on right) and where they received their doctoral degrees (on left). Interactive versions of these figures can be found at: https://dantyrr.github.io/K99-R00-analysis/HBCU_deg.html for **A** and https://dantyrr.github.io/K99-R00-analysis/HBCU_faculty.html for **B**. Source data for *Figure 7* are from Wapman et al. and are available on Zenodo at https://doi.org/10.5281/zenodo.6941651 (*Wapman et al., 2022*).

The online version of this article includes the following source data and figure supplement(s) for figure 7:

**Figure supplement 1.** Pie charts showing where faculty from biology, biomedical, health, and related fields from 32 select institutions received their doctoral degrees (from 2010 to 2020) by percentage.

**Figure supplement 1—source data 1.** Source data of where faculty members received their terminal doctoral degree from 2010 to 2020.

**Figure supplement 1—source data 2.** Source data of the institution type of doctoral degree for faculty at each institution from 2010 to 2020.

**Figure supplement 2.** Pie charts showing the type of institution that faculty from biology, biomedical, health, and related fields from 32 select institutions received their doctoral degrees (from 2010 to 2020) by percentage.

are dominated primarily by faculty with doctoral degrees from other HBCUs and public universities, primarily outside of the top 25 NIH funding category. These data highlight stark differences in hiring practices between HBCUs and institutions that hire significant numbers of R00 awardees. We quantified where faculty at a subset of institutions received their doctoral degrees with annotations for the top 15 institutions (*Figure 7—figure supplement 1*).

## Self-hiring and regional hiring of faculty is ubiquitous

Some institutions tend to hire more faculty with doctoral degrees from the same institution (i.e., self-hires). A quarter of faculty at USC have USC doctoral degrees (*Figure 7—figure supplement 1*). In contrast, only 10–12% of faculty at Duke and Yale have doctoral degrees from Duke and Yale (*Figure 7—figure supplement 1*). In contrast, at Princeton, there are relatively few faculty with Princeton doctoral degrees. Certain institutions (Harvard, Columbia, U Penn, Stanford, Johns Hopkins, University of Michigan, UC Berkeley, Duke, and UNC) are nearly ubiquitous in the top

15 doctoral degrees for faculty at many institutions. There is also regional bias where institutions in the same region or city are more heavily represented. For example, doctoral degree holders from Tufts and Boston University are common for Harvard Faculty. A significant number of faculty at University of Michigan have doctoral degrees from Wayne State University, which is in Detroit, Michigan. The faculty at southeastern institutions have greater representation from Vanderbilt, Emory, Duke, University of North Carolina, and Washington University in St. Louis compared with the rest of the country. Similarly, the University of California system institutions have greater representations from other UC system affiliates including UC Berkeley, UCSD, UCSF, UCLA, USC, and Caltech (*Figure 7—figure supplement 1*).

## Institution faculty hiring is consistent with institution type

We also examined the type of institutions that faculty received their doctoral degrees at (see *Figure 3*). Massachusetts General Hospital, Yale, and Stanford are private institutions with large NIH funding portfolios along with public institutions like UCSF and U of Michigan. We compare these with historically Black colleges and universities and foreign institutions. The faculty at private institutions with large NIH funding portfolios tend to have degrees from other well-funded private institutions (*Figure 7—figure supplement 2*). The faculty at public institutions with large NIH funding portfolios (i.e., UCSF and University of Michigan) tend to have doctoral degrees from other public institutions with large NIH funding portfolios. Similarly, faculty at public institutions with smaller NIH funding portfolios have degrees from other public institutions with smaller NIH funding portfolios (*Figure 7—figure supplement 2*). The faculty at HBCU's tend to have doctoral degrees from public institutions with smaller NIH funding portfolios as well as other HBCUs (*Figure 7—figure supplement 2*). Faculty members with doctoral degrees from HBCUs are rare and do not exceed more than 2% of the total faculty except at other HBCUs and Middle Tennessee State University which has 3.9%. In general, the practice of self-hiring contributes to institutions having faculty with the same doctoral degree type as the institution they are employed at.

## Discussion

It has previously been shown that a minority of institutions populate the faculty ranks of the bulk of other institutions along prestige hierarchies across fields and disciplines (*Clauset et al., 2015*; *Wapman et al., 2022*). These findings are in-line with the Matthew effect whereby success early on increases the chance of success in the future (*Merton, 1968*; *Bol et al., 2018*). Our analyses focus on the K99/R00 career development award. We demonstrate there are significant disadvantages associated with mobility, gender identity, type of R00 institution, and whether faculty candidates move from the K99 institution or stay at the same institution. In the context of the Matthew effect, there are significant advantages for receiving the K99 award at a highly funded institution or for being classified as a man or having mobility in the choice of an R00 institution. Scientists are often the most mobile at the earliest stages of their careers whether by choice or necessity (*Vaccario et al., 2021*). Academic mobility, especially in early stage investigators has been described as 'coerced movement from job to job' and has also been termed 'forced mobility' (*Ackers, 2008*). International postdoctoral fellows may have a cultural reasons for moving to the United States based in perceived bias regarding the United States' scientific training (*Cantwell, 2011*). Our analysis also demonstrates that K99 to R00 faculty flow generally follows similar hierarchies with some deviation for specific institutions. We find that a minority of private and public institutions with the largest NIH funding portfolios produce the majority of K99 awardees, and there are a handful of private and public institutions with the largest NIH funding portfolios that recruit the most R00 awardees. Many K99 awardees from institutions with the largest NIH funding portfolios move to public institutions and institutions with smaller NIH funding portfolios for the R00 phase of the award. These also happen to be less concentrated on the East and West coast. This supports the universal core-periphery structure of hiring that Wapman et al. conceptualized but suggests that prestige hierarchies in academic biomedical science align with NIH funding portfolio size (*Wapman et al., 2022*). Despite this, there are also K99/R00 awardees that move up the prestige hierarchy to public and private institutions with the largest NIH funding portfolios and some individuals that stay within the same category for both K99 and R00 award phases. There are stark differences in future major NIH grant funding depending on several factors including NIH funding

portfolio size of R00 institution, gender identity, and whether the awardee moved or stayed at the K99 institution.

Notably, whether the K99 institution was private or public or the size of the NIH funding portfolio did not impact future funding success except when considering whether the candidate eventually moved for the R00 phase of the award. One key component of activating the R00 award is to demonstrate independence from the candidate's postdoctoral advisor because the candidate will need to establish independence from the postdoctoral advisor to be competitive to receive future NIH funding. Moving to a new institution clearly establishes independence; however, there are many pressures to stay at the K99 institution including familiarity with the institution, location, and family. K99/R00 awardees that stay at the same institution have an increase in the median time to future major NIH funding (either R01, DP2, or R35 MIRA) and have a reduced overall chance to receive any major NIH award. This may be due to reduced start-up packages for internal candidates, which could leave them less well suited to perform the necessary research to be competitive for R01 funding. Candidates that remain at the same institution may also have additional duties in service work that drain their time. They may have more work on committees given the knowledge, experience, familiarity, and institutional history. Alternatively, this may be an internal bias of candidates within the institution or of grant reviewers who do not view the candidate as truly independent from their postdoctoral advisor. There are also cultural biases that largely reward white and Asian male academics by perceiving them to be more productive and de-valuing work done by non-white and non-Asian men (*Blair-Loy, 2023*; *Blair-Loy et al., 2023*). These data bring to light several factors that affect the scientific human experience. Moving institutions throughout one's academic career can cause significant hardship on anyone. First-generation scientists and those in systematically marginalized groups may lack support systems that those from non-marginalized groups have. Familial/friendships ties play important roles in the happiness, support, and productivity in the lives of scientists. Compounding the burden of first-generation scientists and systemically marginalized scientists with the burden of moving away from supports offered by family and friends may have a significant negative impact on those scientists. The second is that obtaining K99 funding and staying at the same institution comes with inherent biases within the system designed to support scientific talent. How these two factors are at odds with each other is of significant note to the wider scientific community. Ameliorating these disparities to increase equity should be of future focus.

Additional disadvantages to future funding success of K99/R00 awardees is whether the candidate is a man or a woman and whether the candidate is from an institution with a large NIH funding portfolio. Each of these factors can prolong the median time to future funding by about half a year and reduces the overall chance of securing future funding by ~5.5%. These cumulative disadvantages may contribute to lower likelihood of receiving promotion and tenure and may contribute to the gender disparities in faculty ranks and in leadership positions. The United States has a more even gender balance for tenure-track positions than some other countries including Japan and South Korea (*Xu et al., 2018*); however, it is apparent that women are at a disadvantage compared with men despite this. There may also be additional disadvantages for K99/R00 awardees who identify as transgender, gender non-binary, or non-conforming, or for individuals from systematically marginalized groups, or those with disabilities. There have been recommendations to promote a sustainable biomedical research enterprise (*Pickett et al., 2015*). Future work is needed to examine how intersectional inequalities impact scientific success within the academic workforce.

We can speculate why K99 awards most often go to individuals at institutions with the largest NIH funding portfolios; however, data on individual K99/R00 applications that were not funded are not publicly available. Our data show that 772 scientists with funded K99 awards did not get R00 awards and 302 of these were from 2019 or earlier. Those made from 2020 onward may still have the possibility to convert to the R00 phase, even after the 2-year K99 period with a 1-year no-cost extension. The factors that prevented the other 302 K99 awardees from 2019 and earlier unable to convert their K99–R00 grants is cause for concern within our greater academic community. Possible explanations include leaving the biomedical workforce, accepting tenure-track positions or other positions abroad, or by simply not successfully securing a tenable tenure-track offer. Of the funded K99/R00 awardees, we can make the assumption that candidates at institutions with large biomedical research portfolios and built-in support to write competitive grant applications that may be lacking at institutions with smaller research enterprises. Institutions with the largest NIH funding portfolios may facilitate greater

access to making scientific discoveries. Movement of postdocs from these institutions to independent faculty positions at institutions with the smaller NIH funding portfolios, may increase the effectiveness of the faculty at those institutions. We must also consider that senior faculty themselves review and score K99/R00 award applications, and by doing so, determine which candidates ultimately receive these awards. A contrary perspective is that K99/R00 awardees at institutions perceived as more prestigious receive their awards in part because of the name and reputation of their institution. Thus, the interpretation of our data is that the system of reviewing and scoring grants has inherent bias toward less prestigious institutions which is currently being reviewed at the NIH. We can extend this speculation of bias to explain why men who receive R00 awards and move to institutions with the largest NIH funding portfolios are the most successful at securing subsequent major NIH awards in contrast to women, people who stay at the same institution for the R00 phase, and individuals at institutions with smaller NIH funding portfolios.

Of note, no K99 award has been made to a candidate at an HBCU, and only two awardees activated R00 awards at HBCUs. Most HBCUs specialize in undergraduate education; however, several perform biomedical research. Thus, we used a different dataset of faculty institutions and doctoral degree institutions to examine how faculty hiring practices differ between institution types, in particular, HBCUs. In agreement with others, we also found that self-hiring is common ( *Altbach et al., 2015*). The self-hiring rate generally between 10% and 20% most institutions including HBCUs. One stark difference between HBCUs and all other US research institutions is that HBCUs appear to be the only institution type that hires faculty with doctoral degrees from other HBCUs. For example, of the 938 biomedical faculty members at Harvard, only 1 has a doctoral degree from an HBCU. Some institutions have lower rates of self-hiring. For example, Princeton faculty have degrees primarily from other Ivy league institutions but doctoral degrees from Princeton only represent the 14th most common for biomedical sciences. In agreement with the known and described systemic inequalities within the academic system we have highlighted in this article, self-hiring within HBCUs may occurs to enable survival of these institutions. Although many HBCUs focus primarily on undergraduate education, several have large biomedical research enterprises, and it is shocking that no K99 awardees have ever been from HBCUs. The K99 MOSAIC program, established in 2020 may increase equity; however, an effort to fund career development awards specifically at HBCUs may also increase equity in this area.

We do not know how many unfunded K99 applications may have originated from HBCUs. A freedom of information act request in 2018 by Dr. Pickett (request # 47950) for fiscal years 2007–2017 for K99/R00 awardees found that the successful award rate for white K99 applicants was 31.0% which was higher than both Asian (26.7%) and Black (16.2%) applicants. Total successful R00 transition rates were also higher among white K99 applicants (77%) compared to Asian (76.1%) and Black (60.0%) awardees (*United States, 2018*). Although data on specific applicants are not available to determine trajectories from specific institution types, these data demonstrate that while improvements in equity to male and female applicants have improved; there is still a significant racial equity gap.

These analyses exclude those that received a K99 but were either unable to transition to an R00 award for various reasons including receiving a faculty position in the United States without needing the R00, receiving a faculty position outside of the United States, pursuing another line of work, or other unknown reasons. Furthermore, these analyses are limited to academic institutions in the United States and do not examine academic centers in other countries. When interpreting these data, it is important to consider that each data point corresponds to an individual with unique motivations for choice of postdoctoral research advisor, institution, and laboratory as well as autonomy in whether to pursue a faculty position and where that may be. Complex consideration of start-up package, institutional clout, and individual motivations factor into this decision. In addition, the choice to give K99/R00 awards and subsequent NIH awards lies mostly with grant reviewers who are generally mid- to senior-career faculty that may have different levels of conscious or unconscious bias for or against various factors that contribute to the overall score given. This dataset does not account for the motivation behind these complex decisions, and an understanding of these motivating factors would be enlightening.

It is important to note the limitations of this study. The present study is focuses on K99/R00 awardees and is limited to faculty flows within the United States. While similar practices may exist in other countries, there are likely significant differences also. Another limitation due to our focus on K99/R00 awardees, is that scientists without K99/R00 awards are excluded from our analyses. Furthermore,

the public data on K99/R00 awardees do not provide demographic details including self-identified gender, race, ethnicity, age, and degree type. The gender identity used here was assigned as a binary man or woman gender based on the first name of the candidate. If these demographic data were available, a critical analysis of bias and how these impact outcomes could be made. For our analysis of K99 awardees, we do not have information on the doctoral degree granting institution, and for our exploration of where faculty members at HBCUs received their doctoral degrees, we do not have information on their postdoctoral research experience. If we had these details, we could make a more complete examination of flow from doctoral degree institution to postdoctoral institution to faculty institution and then to determine outcomes based on each of these. Future research should aim to complete a more comprehensive assessment of faculty flows and outcomes.

Here, we demonstrate the complex nature of faculty hiring within the NIH grant system that exists today in the biomedical sciences and related fields. The K99/R00 award mechanism undoubtedly increases awardees' chance of securing a faculty position. As one of the most coveted NIH grants for postdocs, we have examined the flow of K99 awardees to R00 institutions and how these flows impact future NIH award funding. Many factors besides those considered and quantified here contribute to where K99 awardees choose to begin their faculty careers, and the K99 award itself does not guarantee securing a faculty position. Despite this, we have identified factors that pose significant disadvantages to future funding success for K99/R00 awardees which likely influence funding success more broadly. Future work must examine the role of ethnic and racial bias in these domains. As the K99 MOSAIC program becomes more established, a comparison of this program and the K99 grant mechanism explored here may reveal whether the MOSAIC strategy is effective at promoting equity for underrepresented minorities in the biomedical faculty ranks. By quantifying and understanding these factors, grant reviewers, faculty hiring committees, department chairs, and funding bodies may be able to more equitably award and administer grants and evaluate faculty candidates. Improvements to the faculty hiring process can be achieved centrally through the NIH creating a more equitable grant funding system or within the many diverse faculty searches (*Bhalla, 2019*; *Schmidt et al., 2021*). There is only one NIH and many searches, so systemic change will likely lead to more impactful change; however, both options could be implemented simultaneously for the optimal benefit of the scientific community.

## Materials and methods
### Data acquisition and analysis

Raw K99/R00 data examined in this manuscript are publicly available from the NIH Reporter (https://reporter.nih.gov/). Filtered data and annotated data used to generate *Figures 1–6* and *Tables 1–4* are available in the supplementary files. Names have been removed; however, the corresponding author will provide the identified data upon reasonable request. All data for K99 awardees, R00 awardees, and matched K99/R00 awardees are included in *Supplementary files 1–3*. The data used to generate *Figure 7* and *Figure 7—figure supplements 1 and 2* were generated by Wapman et al. and are available on Zenodo at https://doi.org/10.5281/zenodo.6941651 (*Wapman et al., 2022*). K99 and R00 data were downloaded from NIH Reporter using the 'Advanced Search' function to include only the year range (2006–2022) and only the grant of interest (i.e., K99 or R00). Duplicate values containing additional awards for the same contact PI (i.e., supplemental awards or multiple years of the same award) were removed. The latest available fiscal year for the K99 award was used and the earliest available fiscal year for the R00 was used. The K99 and R00 institutions were aligned and confirmed that the contact PI name was identical before removing the contact PI name from the dataset. The institutions were analyzed using Microsoft Excel and Graphpad Prism (version 9.4.1). Data on doctoral degree university and subsequent faculty position are from *Wapman et al., 2022* and are available at https://larremorelab.github.io/us-faculty/ (*Wapman et al., 2022*). We included the following fields from this dataset: Anatomy, Animal Sciences, Biochemistry, Biological Sciences, Biomedical Engineering, Biophysics, Biostatistics, Environmental Health Sciences, Exercise Science, Kinesiology, Rehab, Health, Health, Physical Education, Recreation, Human Development and Family Sciences, Immunology, Microbiology, Molecular Biology, Neuroscience, Nursing, Nutrition Sciences, Pathology, Pharmaceutical Sciences, Pharmacology, Pharmacy, Physiology, Psychology, Public Health, and Veterinary Medical Sciences. We examined the Historically Black Colleges and Universities from

this dataset. The list of K99 institutions and R00 institutions and HBCU institutions were plotted as Sankey diagrams using the 'gvisSankey()' function in the 'googleVis' package in R (version 2022.07.1, Build 554). Interactive versions of the K99 to R00 Sankey plot are available at: https://k99tor00.shinyapps.io/K99-R00_Sankey/ with data available here: https://zenodo.org/records/10005359 and code available here: https://github.com/chsolis/K99toR00SankeyNetwork2007-2022; (copy archived at *Solis, 2023*). Interactive versions of the HBCU Sankey plots (*Figure 7*) are available at: https://dantyrr.github.io/K99-R00-analysis/ (*Tyrrell, 2023*). The K99 and R00 location maps were generated on ObservableHQ using a map layer and overlaying heatmap dot plot by latitude and longitude coordinates from the NIH Reporter dataset. Data on R01, DP2, and R35 awards were exported from NIH Reporter using the list of K99 to R00 transfers for each year from 2007 to 2023 and the time from each candidate's first R00 budget start date to R01, DP2, and R35 budget start date was calculated.

### Institutional classification

Institutions were classified as either 'highest NIH funding' or 'lower NIH funding' based on 2022 NIH funding levels and included the 25 institutions with the most NIH funding. The institutions included in this list in order from most funding to least are: Johns Hopkins University, University of California, San Francisco, University of Pittsburgh at Pittsburgh, Duke University, University of Pennsylvania, Stanford University, University of Michigan at Ann Arbor, Washington University, Columbia University Health Sciences, University of California, San Diego, University of California Los Angeles, Yale University, University of Washington, Univ of North Carolina Chapel Hill, Massachusetts General Hospital, Emory University, Icahn School of Medicine at Mount Sinai, University of Minnesota, Fred Hutchinson Cancer Center, Northwestern University at Chicago, Brigham and Women's Hospital, New York University School of Medicine, Vanderbilt University Medical Center, and University of Wisconsin-Madison. Leidos Biomedical Research, Inc is also included in this list but does not have any K99 or R00 awardees.

### Binary name-to-gender classification

Name-based ascription tools for gender and for race/ethnicity are inherently biased and these biases occur unevenly across groups (*Lockhart et al., 2023*). A name-to-gender assignment tool (GenderAPI, https://gender-api.com/) was used to assign a binary gender (man/woman) to K99/R00 awardees. GenderAPI was chosen based on its low rate of mis- and non-classifications (*Santamaría and Mihaljević, 2018*). It assigns gender identity as either men, women, or unknown. Because the terms male and female typically refer to biological sex, the terms man/men or woman/women are utilized here. However, it is important to note that gender identity is a social construct that is non-binary and non-static. Individuals identifying as transgender, non-binary, or gender non-conforming, or whose gender identity has changed since the time of receiving the K99 or R00 award are at risk of being misgendered. We acknowledge this remains a major limitation of this type of name-to-gender methodology and we hope it changes in the future to be more inclusive.

Cox proportional hazard model: Cox proportional hazard model was calculated using the subset of data with matched K99 and R00 awards from *Supplementary file 3*. Specifically, we examined whether individuals received an R01, DP2, R35 award. We calculated the time from the start of the R00 budget until the start of the first R01, DP2, or R35 award and determined how binary gender, mobility, private/public institution, and institutional NIH funding amount affected the chance and time to receive subsequent R01, DP2, or R35 award. These data were analyzed using RStudio (version 2022.01.1 Build 554) with R (version Spotted Wakerobin, Release 7872775e, 2022-07-22 for macOS) using the 'survfit()', 'coxph()', and 'autoplot()' functions.

## Acknowledgements

We thank Chris Pickett for his valuable input when preparing this publication. CS is supported by NIH R00-HL151825. TV is supported by NIH F32-GM140568. DJT is supported by NIH R00-AG068309. SRH is supported by NIH R00-ES033738.

## Additional information

### Funding

| Funder | Grant reference number | Author |
|---|---|---|
| National Institutes of Health | R00AG068309 | Daniel J Tyrrell |
| National Institutes of Health | R00ES033738 | Sarah R Hengel |
| National Institutes of Health | HL151825 | Christopher Solis |

The funders had no role in study design, data collection, and interpretation, or the decision to submit the work for publication.

### Author contributions

Nicole C Woitowich, Joel Babdor, Data curation, Formal analysis, Investigation, Methodology, Writing - review and editing; Sarah R Hengel, Data curation, Formal analysis, Funding acquisition, Investigation, Methodology, Writing - review and editing; Christopher Solis, Data curation, Software, Formal analysis, Funding acquisition, Investigation, Methodology, Writing - review and editing; Tauras P Vilgalys, Data curation, Formal analysis, Investigation, Visualization, Methodology, Writing - review and editing; Daniel J Tyrrell, Conceptualization, Data curation, Software, Formal analysis, Funding acquisition, Investigation, Visualization, Methodology, Writing - original draft, Writing - review and editing

### Author ORCIDs

Nicole C Woitowich ⓘD http://orcid.org/0000-0002-3449-2547
Sarah R Hengel ⓘD http://orcid.org/0000-0002-0408-8623
Daniel J Tyrrell ⓘD http://orcid.org/0000-0002-0811-6724

Reviewer #1 (Public Review): https://doi.org/10.7554/eLife.88984.4.sa1
Reviewer #2 (Public Review): https://doi.org/10.7554/eLife.88984.4.sa2
Reviewer #3 (Public Review): https://doi.org/10.7554/eLife.88984.4.sa3
Author Response https://doi.org/10.7554/eLife.88984.4.sa4

## Additional files

### Supplementary files

• Supplementary file 1. Curated NIH Reporter dataset of K99 recipients from 2006 to 2022.

• Supplementary file 2. Curated NIH Reporter dataset of R00 recipients from 2006 to 2022.

• Supplementary file 3. Curated NIH Reporter dataset of matched K99 and R00 recipients from 2006 to 2022.

• MDAR checklist

### Data availability

All data generated or analyzed during this study are included in the manuscript and supporting files.

The following dataset was generated:

| Author(s) | Year | Dataset title | Dataset URL | Database and Identifier |
|---|---|---|---|---|
| Solis C | 2023 | chsolis/K99toR00SankeyNetwork2007-2022: K99 to R00 Sankey Network App v1.0 | https://doi.org/10.5281/zenodo.10005359 | Zenodo, 10.5281/zenodo.10005359 |

The following previously published dataset was used:

| Author(s) | Year | Dataset title | Dataset URL | Database and Identifier |
|---|---|---|---|---|
| Wapman KH, Zhang S, Clauset A, Larremore D | 2022 | Quantifying hierarchy and dynamics in U.S. faculty hiring and retention | https://doi.org/10.5281/zenodo.6941651 | Zenodo, 10.5281/zenodo.6941651 |

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
