## [Editor Report · eLife assessment]

This study follows the career trajectories of the winners of an early-career funding award in the United States, and finds that researchers with greater mobility, men, and those hired at well-funded institutions experience greater subsequent funding success. Using data on K99/R00 awards from the National Institutes of Health's grants management database, the authors provide **compelling** evidence documenting the inequalities that shape faculty funding opportunities and career pathways, and show that these inequalities disproportionately impact women and faculty working at particular institutions, including historically black colleges and universities. Overall, the article is an **important** addition to the literature examining inequality in biomedical research in the United States.

---

## [Referee Report · Reviewer #1 (Public Review)]

Summary and strengths:

This is an interesting, timely and informative article. The authors used publicly available data (made available by a funding agency) to examine some of the academic characteristics of the individuals recipients of the National Institutes of Health (NIH) k99/R00 award program during the entire history of this funding mechanism (17 years, total ~ 4 billion US dollars (annual investment of ~230 million USD)). The analysis focuses on the pedigree and the NIH funding portfolio of the institutions hosting the k99 awardees as postdoctoral researchers and the institutions hiring these individuals. The authors also analyze the data by gender, by whether the R00 portion of the awards eventually gets activated and based on whether the awardees stayed/were hired as faculty at their k99 (postdoctoral) host institution or moved elsewhere. The authors further sought to examine the rates of funding for those in systematically marginalized groups by analyzing the patterns of receiving k99 awards and hiring k99 awardees at historically black colleges and universities.

The goals and analysis are reasonable and the limitations of the data are described adequately. It is worth noting that some of the observed funding and hiring traits are in line with the Matthew effect in science (Merton, 1968: https://www.science.org/doi/10.1126/science.159.3810.56) and in science funding (Bol et al., 2018: https://www.pnas.org/doi/10.1073/pnas.1719557115). Overall, the article is a valuable addition to the research culture literature examining the academic funding and hiring traits in the United States. The findings can provide further insights for the leadership at funding and hiring institutions and science policy makers for individual and large-scale improvements that can benefit the scientific community.

Weaknesses:

The authors have addressed my recommendations in the previous review round in a satisfactory way.

---

## [Referee Report · Reviewer #2 (Public Review)]

Summary and strengths:

Early career funding success has an immense impact on later funding success and faculty persistence, as evidenced by well-documented "rich-get-richer" or "Matthew effect" phenomena in science (e.g., Bol et al., 2018, PNAS). In this study the authors examined publicly available data on the distribution of the National Institutes of Health's K99/R00 awards - an early career postdoc-to-faculty transition funding mechanism - and showed that although 89% of K99 awardees successfully transitioned into faculty, disparities in subsequent R01 grant obtainment emerged along three characteristics: researcher mobility, gender, and institution. Men who moved to a top-25 NIH funded institution in their postdoc-to-faculty transition experienced the shortest median time to receiving a R01 award, 4.6 years, in contrast to the median 7.4 years for women working at less well-funded schools who remained at their postdoc institutions.

Amongst the three characteristics, the finding that researcher mobility has the largest effect on subsequent funding success is key and novel. Other data supplement this finding: for example, although the total number of R00 awards has increased, most of this increase is for awards to individuals moving to different institutions. In 2010, 60% of R00 awards were activated at different institutions compared to 80% in 2022. These findings enhance previous work on the relationship between mobility and ones' access to resources, collaborators, or research objects (e.g., Sugimoto and Larivière, 2023, Equity for Women in Science (Harvard University Press)).

These results empirically demonstrate that even after receiving a prestigious early career grant, researchers with less mobility belonging to disadvantaged groups at less-resourced institutions continue to experience barriers that delay them from receiving their next major grant. This result has important policy implications aimed at reducing funding disparities - mainly that interventions that focus solely on early career or early stage investigator funding alone will not achieve the desired outcome of improving faculty diversity.

The authors also highlight two incredible facts: No postdoc at a historically Black college or university (HBCU) has been awarded a K99 since the program's launch. And out of all 2,847 R00 awards given thus far, only two have been made to faculty at HBCUs. Given the track record of HBCUs for improving diversity in STEM contexts, this distribution of awards is a massive oversight that demands attention.

At no fault of the authors, the analysis is limited to only examining K99 awardees and not those who applied but did not receive the award. This limitation is solely due to the lack of data made publicly available by the NIH. If this data were available, this study would have been able to compare the trajectory of winners versus losers and therefore could potentially quantify the impact of the award itself on later funding success, much like the landmark paper by Bol et al. (PNAS; 2018) that followed the careers of an early career grant scheme in the Netherlands. Such an analysis would also provide new insights that would inform policy.

Although data on applications versus awards for the K99/R00 mechanism are limited, there exists data for applicant race and ethnicity for the 2007-2017 period, which were made available by a Freedom of Information Act request through the now defunct Rescuing Biomedical Research Initiative (https://web.archive.org/web/20180723171128/http://rescuingbiomedicalresearch.org/blog/examining-distribution-k99r00-awards-race/). These results are highly relevant given the discussion of K99 award impacts on the sociodemographic composition of U.S. biomedical faculty. During the 2007-2017 period, the K99 award rate for white applicants was 31% compared to 26.7% for Asian applicants and 16.2% for Black applicants. In terms of award totals, these funding rates amount to 1,384 awards to white applicants, 610 to Asian applicants, and 25 to Black applicants. However, the work required to include these data may be beyond the scope of the study.

The conclusions are well-supported by the data, and limitations of the data and the name-gender matching algorithm are described satisfactorily.

---

## [Referee Report · Reviewer #3 (Public Review)]

Summary:

The researchers aim add to the literature on faculty career pathways with particular attention to how gender disparities persist in the career and funding opportunities of researchers. The researchers also examine aspects of institutional prestige that can further amplify funding and career disparities. While some factors about individuals' pathways to faculty lines are known, including the prospects of certain K award recipients, the current study provides the only known examination of the K99/R00 awardees and their pathways.

Strengths:

The authors establish a clear overview of the institutional locations of K99 and R00 awardees and the pathways for K99-to-R00 researchers and the gendered and institutional patterns of such pathways. For example, there's a clear institutional hierarchy of hiring for K99/R00 researchers that echo previous research on the rigid faculty hiring networks across fields, and a pivotal difference in the time between awards that can impact faculty careers. Moreover, there's regional clusters of hiring in certain parts of the US where multiple research universities are located. Moreover, documenting the pathways of HBCU faculty is an important extension of the study by Wapman et al. (2022: https://www.nature.com/articles/s41586-022-05222-x), and provides a more nuanced look at the pathways of faculty beyond the oft-discussed high status institutions. (However, there is a need for more refinement in this segment of the analyses). Also, the authors provide important caveats throughout the manuscript about the study's findings that show careful attention to the complexity of these patterns and attempting to limit misinterpretations of readers.

Weaknesses:

The authors have addressed my recommendations in the previous review round in a satisfactory way.

---

## [Author Response]

The following is the authors’ response to the previous reviews.

**Recommendations for the authors:**

**Reviewer #1 (Recommendations for the Authors):**
The authors have addressed my recommendations in the previous review round in a satisfactory way. I only have one additional comment to the authors:In the manuscript abstract lines 31-32, the author state that: "Using NIH data for the period 2006-2022, we report that ~230 K99 awards were made every year, representing ~$25 million annually."-- The "~$25 million" is under-stating the actual funds spent because this sum is just money spent on the first year of some k99s while the NIH is paying years 2,3,4 etc for others for k99 awards (~90% conversion rate to R00) awarded in previous years for a given year. The NIH is actually spending ~$230-$250 million a year on the k99 award mechanism in a given year. so the authors need to amend the stated amount in the manuscript.

Thank you for pointing this out. The reviewer is correct, that we had incorrectly only calculated the investment $ in new K99 awards made. We have corrected this in the revised manuscript. We appreciate your careful reading of our manuscript and the edits made based on your comments have improved the final version.

**Reviewer #2 (Recommendations for the Authors):**
Thank you for taking the time to revise this important work. I learned a lot reading this paper a second time, and appreciate the improvements you have made.My only major thought while re-reading this is that I wish you all had written two papers! I see two themes in this work: one looking at faculty hiring networks from the Wapman et al. dataset, and another at K99/R00 conversions by institution, gender, and researcher mobility and its impact on subsequent funding success. After reading, I felt like I had many follow-up questions about both analyses, but it would be impractical for me to suggest all these follow-up analyses without making your paper unreasonably long.

Thank you for these comments. We agree that there are 2 general themes in this paper. While we feel that significantly expanding on both themes will be important in future research. Our hope is that this work continues to inspire others to critically examine funding practices and inequity in the same way that the work of Wapman, Pickett, etc. inspired the present work.

For example, regarding the results that more R00 are activated at different institutions, and that moving institutions improves subsequent funding success, I wonder: Do proportionally more women or men move institutions? Do proportionally more K99 awardees at less-funded places move for their R00, or less? The Cox proportional hazard models illustrate the impact of various characteristics on subsequent funding success, but they do not illustrate disparate impacts of mobility on different groups (if I am understanding them correctly). (You sort of dive into these questions in the very interesting subsection, "K99/R00 awardee self-hires are more common at institutions with top NIH funding." I wanted to read more!)

Thank you for these kind comments. These are fantastic follow-up questions. We do not feel that we can adequately address them within the present manuscript without potentially splitting it into 2 separate manuscripts. However, we may examine these in future analyses. We are particularly interested in examining additional aspects such as how the K99 MOSAIC funding mechanism may differ from the traditional K99 mechanism. Since the K99 MOSAIC mechanism is newer, there may not be enough K99 MOSAIC awards made for a thorough exploration.

As another example, for your analysis on faculty hiring networks, the prevalence of self-hiring amongst institutions and regions was one finding. However, this finding seems somewhat at odds with the previous takeaway about how researcher mobility improves subsequent funding success. Are institutions doing themselves a disfavor by hiring their own, then? I suspect there is more to say here about this pattern... maybe there are important differences between PhD institution and postdoc institution and its impact on hiring/subsequent funding success? Or is this a story about upward mobility into the top 25 well-funded NIH institutions?

Again, these are very insightful comments and follow-up questions. We hope to address these in potential future manuscripts. We also hope that others may become interested in finding answers to these questions by exploring our dataset as well as other publicly available datasets such as the Wapman et al. dataset.

I can completely understand how combining the faculty hiring network analysis with the K99/R00 conversions would seem like a natural fit, but I personally feel - emphasis on this being a *personal* opinion - that there would have been benefits to giving more space to the details of both analyses separately. Perhaps this is a "hindsight is 20/20" issue. Or an issue with the current times in which ones' brain can only hold so many main takeaways from a single body of work. (For example, I struggled to summarize your paper in my public review because I find so many takeaways important.)I suppose this is all to say that I find your work important enough to warrant additional follow-up work! :

Thank you for these very kind remarks. This work evolved over 8-10 months as evidenced by the updates to the biorXiv preprint. With unlimited time and foresight, it would probably be best to have separated the 2 themes into separate manuscripts and expanded both. Given current constraints, we plan to make some changes/updates to the present manuscript and hopefully include more in-depth analyses on each theme in future works. Thank you again for the thoughtful reading and critique of both our original manuscript and the revised version.

Minor comments/questions:"K99 to R00 conversions are increasing in time"Assuming I am interpreting the figures correctly, in my opinion, the most important takeaway is that *the number of R00 awards have increased, but only for awardees moving to another institution.* This key result, best illustrated by panels A and C of Figure 1, is buried in the long paragraph in this section. The organization of content in this section could be improved and more focused. Consider renaming this subsection to be more declarative: "K99 tR00 conversions have increased, but only for awardees moving to another institution."

This is a very concise interpretation of this data. We have edited the paragraph referenced by the reviewer, split it into 2 paragraphs, and changed the title to “K99 awardees increasingly move to other institutions for R00 awards from 2008 to 2022” and the final sentence to “Thus, the number of K99 to R00 conversions is consistent over time, but increasingly more R00 awardees have moved to other institutions since 2013”

Similarly, I personally found the current title of the subsection, "K99 to R00 conversions are increasing with time" is mildly confusing. An R00 award indicates a successful conversion, so why not simply call this an R00 award instead of saying K99-to-R00 conversion? Also, when I look at Figure 1B and exclude the conversion rates for 2007 and 2008 (because this is a 3 year rolling average), I see that conversion *rates* (or R00 awards) have remained stagnant. This comment is very much in-the-weeds and is mainly to do with clarity of language.

Thank you for these comments. We had “K99 to R00 conversion” to highlight the unique nature of this award mechanism that a person can only receive an R00 if they previously had a K99 award. Nevertheless, we have edited the text to “R00 awards” and “R00 awardees” to simplify things. We also want to note that we did not compute a 3-year rolling average. The function we used was: (X/(Y -1))x100 where X is the number of R00 awards made in a year and Y is the number of K99 awards made in a year. We did note an error in our calculation in the previous version of the manuscript. Previously, we included all R00 awards and K99 awards for each year from the NIH Reporter dataset; however, this is a flawed methodology. NIH reporter includes only extramural K99 award data and extramural R00 awards, but intramural K99 awardees can receive extramural R00 awards and thus are only included in the R00 dataset. There were 141 R00 awardees in our dataset from NIH Reporter that did not have K99 data, so we assume these are intramural K99 awards since it is required to have a K99 to be eligible for the R00 award. Since we do not know the awarding year for intramural K99 awardees or have data on intramural K99 awardees that fail to activate the R00 award (or stay internal at NIH), we have excluded these 141 R00 awardees. In the previous version, this mis-calculation exaggerated rolling conversion rate (we had correctly calculated the 78% total conversion rate). We re-analyzed our rolling conversion rate and found the average is 81.8% (excluding the first 2 years of the K99 program and the last 2 years).

This is a long explanation, but essentially, we overestimated the number of R00 awards which inadvertently increased the rolling conversion rate. We have corrected this and simplified the first 2 paragraphs of the Results section.

I was also mildly confused looking at Figure 1c. The caption says that the percentages represent the K99 awardees that stayed at the *same* institution for the R00 activation, but the percentages are next to the solid circles which the legend labels as "different institution." Perhaps another or different way to show this is a stacked bar chart, where one bar represents the percentage of R00 awards activated at the same institution and another bar represents the percentage of R00 awards activated at a different institution. The bars always add to 100% but the change in proportions illustrates that proportionally fewer awards are being made to those remaining at the same institution.

Great idea. We have included a stacked bar chart here. Since the stacked bar chart is percentages, we felt it was important to also show the total numbers so we still included the previous chart also but removed the percentage numbers from it. We also changed the departmental analysis to stacked bar charts. This shows the stark difference between 2008-2012 and 2013 onward. These changes were made in the revised Fig. 1.

Minor question: I would love to see Table 3 and Table 4 as a time-series. Has the proportion of recipients at various institution types changed with time?

This is a great suggestion and we felt it fit best in Figure 5, so we’ve added it there.

Table 3 is useful but only indirectly addresses my first "Recommendation to the Authors" from my previous review. I did some number crunching myself from the data provided. Assuming I did this correctly: If you're a K99 awardee at a private institute, you had a 76.3% change of getting an R00 compared to 80.4% for a K99 awardee at a public institution. If you're a K99 awardee at a top-funded institution, you had a 76.8% chance of R00 compared to 78.6% for a lower-funded institution. I would have liked to see more figures and tables to illustrate *conversion rates* by institution type in this way. Interestingly, to me, these data suggest that there are not enormous conversion rate differences by institution type (though looking at these now, I am confused at the 89% statistic in line 174 and where that comes form, since it is much higher than what I've calculated).

Thank you for this suggestion and these comments. Please see above where we describe how we incorrectly overestimated the 89% statistic. This has been corrected. As the reviewer suggested, we now show yearly percent of grants to specific institution types in the revised Figure 5. We agree with the reviewer that showing the conversion rate by institution type is interesting; however, it is fairly obvious from the new panels in Figure 5 that there is not much difference in conversion rate. Thus, to avoid crowding too many panels into the figure, we opted to keep the stacked bar plot.

**Reviewer #3 (Recommendations for the Authors):**
-One minor change to Figure 1C would be to switch the color coding for the lines so that they match with 1D whereby "same institution" would be white circles, or whatever the authors decide would be best for consistency since they are similar comparisons.

Thank you for this suggestion. We have corrected this to be consistent.

-Minor note for lines 459-461: I would suggest changing the wording to "intersectional inequalities" as it is not that a scientist's identities impact their careers as much as how those identities are positioned within an unequal opportunity structure and differentially treated that produce varying career trajectories and experiences of marginalization and cumulative (dis)advantages.

Thank you and we agree with you. We have made this correction.

-To carry forward a suggestion for the authors in my previous review, future research that more fully explores the research infrastructure of institutions for how top NIH funded institutions continue to be top funded institutions year after year could help clarify some of the career mobility and same/similar institution hiring found in the data. Rather than hand coding institutions for some of the infrastructure, the National Center for Education Statistics' Integrated Postsecondary Education Data System (IPEDS) has data on colleges and universities including whether they operate a hospital, have a medical degree, and many other interesting data about student and faculty demographics, institutional expenditures (including research budgets), and degrees awarded in different fields of study (undergrad and grad) that may be helpful to the authors as they continue their research stream in this area.

Thank you very much. We will look into this data set as we continue our investigations in this area.